# On the Performance of Secure Sharing of Classified Threat Intelligence between Multiple Entities

**DOI:** 10.3390/s23020914

**Published:** 2023-01-12

**Authors:** Ricardo Fernandes, Sylwia Bugla, Pedro Pinto, António Pinto

**Affiliations:** 1INESC TEC, 4200-465 Porto, Portugal; 2Instituto Politécnico de Viana do Castelo, 4900-347 Viana do Castelo, Portugal; 3Instituto Universitário da Maia, 4475-690 Maia, Portugal; 4Centro de Inovação e Investigação em Ciências Empresariais e Sistemas de Informação, Escola Superior de Tecnologia e Gestão, Instituto Politécnico do Porto, 4610-156 Felgueiras, Portugal

**Keywords:** performance, threat intelligence, secure sharing

## Abstract

The sharing of cyberthreat information within a community or group of entities is possible due to solutions such as the Malware Information Sharing Platform (MISP). However, the MISP was considered limited if its information was deemed as classified or shared only for a given period of time. A solution using searchable encryption techniques that better control the sharing of information was previously proposed by the same authors. This paper describes a prototype implementation for two key functionalities of the previous solution, considering multiple entities sharing information with each other: the symmetric key generation of a sharing group and the functionality to update a shared index. Moreover, these functionalities are evaluated regarding their performance, and enhancements are proposed to improve the performance of the implementation regarding its execution time. As the main result, the duration of the update process was shortened from around 2922 s to around 302 s, when considering a shared index with 100,000 elements. From the security analysis performed, the implementation can be considered secure, thus confirming the secrecy of the exchanged nonces. The limitations of the current implementation are depicted, and future work is pointed out.

## 1. Introduction

Sharing cyberthreat information among multiple agencies and organizations is an important shield against cyberattacks. Being able to identify, assess, monitor, and respond to cyberthreats reduces the risk exposure of networks, systems, and data. Platforms controlling cyberthreat sharing, such as the Malware Information Sharing Platform (MISP) [1], address security incidents as a community by connecting and sharing information concerning attacks or threats.

The core MISP operation enables the open dissemination of threat or incident-related information within a community where everyone can be a consumer and/or a contributor/producer of the threat intelligence. This information exchange is a data synchronization procedure between different MISP instances through pull/push mechanisms, assuming an existing trust relationship between the organizations exchanging information. In case this relationship does not exist, the MISP presents limitations for organizations wanting to impose stricter control over information disclosure. Thus, the same authors proposed in [2] a controlled information-sharing functionality empowered with a searchable encryption algorithm to impose greater control and confidentiality of the information sharing between the MISP instances. It used Symmetric Searchable Encryption (SSE) which is a searchable encryption that uses symmetric key cryptography to encrypt the search queries. However, the proposed system was not assessed regarding its impact on the MISP performance.

In this paper, the authors:describe the prototype implementation of the previously proposed system;evaluate the performance of the implementation;propose a new index update procedure that boosts the performance.

The prototype implementation of the previously proposed system considers multiple elements exchanging information with each other and evaluates the performance of this system based on a searchable encryption technique. The evaluation takes into account the two key processes, namely the process of generating the symmetric key of a shared group and the process of uploading data to the shared index (the MongoDB collection). The two processes are evaluated on their time duration and Random Access Memory (RAM) and Central Processing Unit (CPU) usage during the process execution. Moreover, this paper proposes and describes an index update process improvement by including a caching system to reduce the duration of the update process.

This paper is structured as follows. Section 2 presents the related work regarding the searchable encryption and SSE performance evaluation. Section 3 details the implementation of the prototype. Section 4 presents the performance evaluation. Section 5 describes the proposed improvements based on the performance results. Section 6 provides a security analysis for the implemented prototype. Section 7 concludes the work herein and references future works.

## 2. Related Work on Searchable Encryption and Performance

Threat intelligence refers to evidence-based knowledge about an existing or potential threat that can help entities in preventing an attack or accelerating the detection of compromised assets [3]. Examples of threat intelligence are the Indicators of Compromise (IoCs); Tactics, Techniques, and Procedures (TTPs); security alerts; threat intelligence reports; tool configurations; and other structured or unstructured data which can be received by an entity from a variety of internal and external sources [4]. A major part of the threat intelligence cycle is the analysis and distribution of credible information and its utilization. The MISP platform is a tool which effectively allows for the storing, dissemination, and sharing of a wide range of cyberthreat intelligence within communities, companies, or organizations. Such open dissemination of threats or incident-related information helps to detect and protect against indicators of compromise and to avoid or mitigate the impact of attacks. The information exchange is treated as a data synchronization procedure between different instances, where the data are usually only encrypted during transport. An approach using searchable encryption was proposed in [2] and used to allow search operations within the shared data while preserving its confidentially and integrity [5].

Searchable encryption is defined as searching for encrypted data located on a server or cloud without the server learning anything from the data [6]. This search procedure assumes an encryption scheme that allows for the sending of a collection of encrypted data to the server while supporting keyword searches on these data [5,7,8,9]. The current solutions for searchable encryption either use an encryption algorithm that allows search operations to be performed on the ciphertext or an index that is created based on existing keywords. Using indexes is more efficient and can increase search performance since they allow queries to be performed using trapdoors generated from the keywords to be searched. A trapdoor can be the result of a cryptographic hash function over the text to be queried. These functions are simple to calculate when in possession of all the data but time-consuming to reverse without knowing the original cleartext information. However, index-based solutions typically require additional calculations, which are performed at the information storage stage, to extract keywords, encrypt them, and then add them to the index. There are two main approaches to building a searchable index: a forward index and an inverted index. A forward index takes the form of a list of keywords per document [5], while an inverted index takes the form of a list of documents by keywords, making it easier to identify all the documents that contain a given keyword [10,11].

In the literature, the dominant techniques of carrying out searchable encryption are (1) SSE and (2) Public Key Encryption with Keyword Search (PEKS).

The SSE uses symmetric key cryptography where only the owner of the secret key can produce ciphertexts and perform searches. Various types of SSE schemes exist, namely (1) with a sequential scan [6], (2) with a secure index [10,12], (3) dynamic schemes [13,14,15], (4) fuzzy keyword search schemes, which tolerate minor typos and formatting inconsistencies in the search queries [16,17], and last but not least, (5) conjunctive [18], ranked [19], or verifiable keyword search schemes, which verify whether the search results are complete and correct [20].

The common problems of using the SSE, identified in the literature, are information leakage [15], the discovery of the cryptographic hashes of keywords contained in an updated document [21], or the inefficiency in terms of search and index update times. Over time, improved solutions for searchable encryption appeared. In [15], the authors introduced the concept of Dynamic Searchable Symmetric Encryption (DSSE), which allows a client to encrypt data in such a way that it can later generate lookup keys and send them as queries to a storage server. Given a token, the server can search over the encrypted data and return the appropriate encrypted files, ensuring a significant reduction in information leakage and maintaining efficiency in search and update operations. The authors in [22] introduced a new solution, called *Blind Storage*. This allows clients to store a set of files on remote servers, such as Dropbox, in such a way that the server cannot know how many files are stored there or their size. The server only knows of the existence of a file when it is obtained by a client, without ever knowing the name of the file or its content. A comprehensive survey was written by Poh et al. in [23], where the authors examined and categorized the existing SSE schemes in the literature. The authors compared four main approaches in designing an SSE scheme, namely without an index, with direct indexing, with inverted indexing, and with tree structures, and analyzed the performance of the searching and updating operations of these four structures. The results of a search operation show that an inverted index and a tree-based index allow for a sublinear search, while a direct index normally results in a linear search.

The PEKS allows anyone to create ciphertexts using a public key, but only the owner of the corresponding private key can perform the searches [24]. This makes the PEKS easy to use in multi-user scenarios. The first PEKS scheme was proposed by Boneh et al. [25], where the authors used identity-based encryption, in which the keyword acts as the identity. The Boneh scheme only supports single keyword retrieval; thus, further various works appeared which proposed PEKS schemes that support multiple keywords queries, such as a disjunctive keyword search [26], conjunctive keyword search [27], and Boolean keyword search [28]. The latter researchers also tackled other aspects of the PEKS, such as the security, precision, and efficiency [29,30,31], and their works have greatly improved the usefulness of the PEKS schemes. Security shortcomings for the PEKS scheme were reported, for example, in [22], where keyword guessing attacks are used to endanger the privacy of the users and the searches they perform.

Wang et al. [5] presented a comparison of several classic SSE and PEKS schemes and also mentioned further works related to searchable encryption which focus on testing the attacks on the schemes, an analysis of the vulnerabilities, and approaches for improving the schemes’ security, functionality, and performance.

The solution proposed in [2] by the same authors of this article addresses the controlled information-sharing functionality of a MISP system and uses a searchable encryption algorithm which is a variation of the SSE. It uses a symmetric key to encrypt search queries. None of the works surveyed is comparable with the solution, as there are specific characteristics not considered before, such as different categories of shared information, time-limited information sharing, and the existence of an external data source such as the MISP system. Moreover, no performance assessment for this solution is available.

A set of research works evaluate the performance cost of solutions using SSE techniques. The efficiency of a scheme is measured by the generated costs associated with the computation produced by the communication performed. A common approach for performing such a performance comparison is using the *Big O* notation, an example being [14]. The base for a comparison of the performance was first defined by Sand et al. [6], with a solution that achieved a search time that is linear in the length of the file collection. The problem with using the *Big O* notation is that it is independent of the implementation and, thus, not applicable to the objective of this work.

## 3. Prototype Implementation

This section details the prototype implementation of the solution proposed in [2] to support multiple elements exchanging information, with a focus on the core functionalities that allow data sharing through a shared index, i.e., the functionality to generate the symmetric key of a sharing group and the functionality to update a shared index.

Sharing data using a shared index allows multiple entities to have a common means of sharing information. A shared index is used in situations where multiple entities agree to form an information-sharing group, defining the permissions to update and search the shared index. A sharing group refers to a mutual connection between two or more entities that decide to share information, where one entity can be part of multiple groups. The creation of a sharing group starts with the naming of the group, guaranteeing its uniqueness. Subsequently, a relation is made with the entities which will be part of the group, with all agreeing on a symmetric key. The relationship between sharing groups and entities is of a many-to-many type, which means that one group can have several entities associated with it and one entity can be associated with several groups. The shared index contains references to the information existing in the MISP instances of each one of the entities forming the group. When an entity performs a search over the indexed data, the result of the search is the identification of the entity that has relevant information about that search. Consequently, the entity can make data synchronization requests to the entities that reported having related information.

### 3.1. Prototype Description

The prototype was built to simulate the exchange of information between two distinct organizations. The implemented prototype is depicted in Figure 1 implementing two instances of the proposed solution, communicating with each other. In Instance 1, the developed REST API is running on the computer as localhost. This API is connected to two components: a MySQL database used in the general operation of the system and a virtual machine running an instance of the MISP platform. In Instance 2, there is a virtual machine running a Linux server running Docker. The developed Rest API and another instance of the MySQL database are executed within the Docker environment. Finally, as in Instance 1, the Rest API of Instance 2 is connected to an instance of the MISP platform. To enable the secure sharing of confidential information through a shared index, a remote MongoDB database was also used to store the indexes created for each sharing group. The components used were the following:An HP computer, model Pavilion 14-ce3014np, running Windows 10 with an Intel(R) Core(TM) i7-1065G7 CPU@1.30 GHz–1.50 GHz and 16 GB of RAM;Two instances of the MySQL 5.7 database;Two VirtualBox (version 6.1.18) machines, each one running a MISP instance (version 2.4);A VirtualBox machine running a Linux server (Ubuntu 18.04.5 LTS) and running a Docker environment (version 20.10.7);A remotely allocated MongoDB (version 4.4.9) database instance.

**Figure 1 sensors-23-00914-f001:**
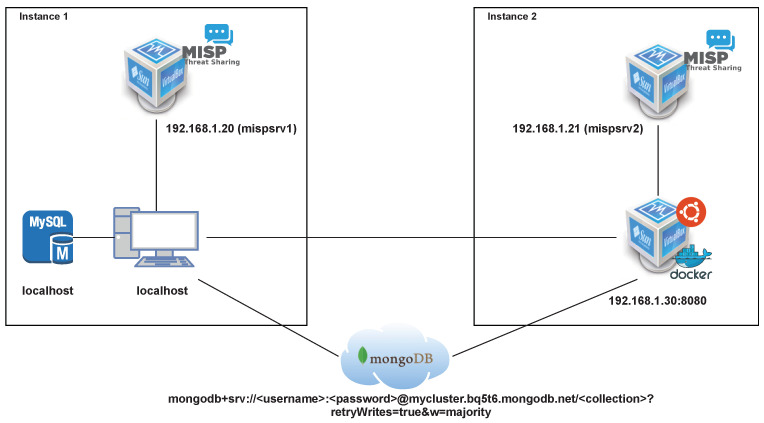
Implemented prototype.

### 3.2. Symmetric Key Generation

Each shared group’s symmetric key is generated with contributions from all entities in the group. The resulting key is used by all the members of the group to manipulate the data existing in the shared index. This symmetric key has two main uses: in the creation of the trapdoors (for both index update and search) and to encrypt the digital signature of the trapdoor that will be inserted in the index. The symmetric key is built in a sequential manner because the entity that decides to initiate the process triggers the remaining entities in the group, causing them to also initiate the key derivation process. At the end of the key derivation, each entity must contain a number of contributions equal to the number of entities present in the group, i.e., its own contribution together with the contribution of the other group entities. After finishing these procedures, the system sorts the contributions (hashes) and applies a hash function to obtain the final symmetric key.

Two preconditions are required for the key construction process to work properly on all entities in the group. The first condition is that all entities create the group with the same name, the group name being unique. The second condition is that the associations between groups and entities must be completed by each member of the group in order to avoid contradictory results in the different entities that compose the group. The algorithm was also prepared to verify that the set of Universally Unique IDentifier (UUID) of the entities is not equal to any entity present in the group.

Listing 1 presents the implementation of the initial key generation request. The first step is to check if there are entities associated with the group (lines 1–4). If there are no entities in the group, it is not possible to send requests and, in this case, the process is cancelled. The next step of the process is the generation of the secure hash that will become the entities’ contribution and will be used in the final symmetric key derivation (line 5). The function *getEntityCurrentHashGroup* receives the group identifier and the timestamp of the moment when the request is made. Before generating a new secure hash, the function checks whether the system has already generated a hash for this group. The generated hashes are saved for reuse, but only for the respective groups. Different groups have different hash contributions. If there is no hash saved, a new hash contribution is generated. The resulting contribution is stored in a temporary list (line 6). This temporary list is responsible for storing all contributions of all entities. After the final key generation, there is no need to store the contributions received from the other entities, so the list is deleted. From this moment on, it is already possible to send contribution requests per entity in the group (lines 7–24). The first step is to build the request (line 8). This request has the same format as any other request made between entities and consists of an entity identifier, a timestamp, and encrypted content. In the current implementation, the encrypted content includes the UUID, a timestamp (Ts), the group name, the list of identifiers of the entities that belong to the group, and the contribution hash. The group name and the list of entity identifiers will be used to guarantee the integrity of all entities that belong to the group. The contribution hash will enable the other entities to build the final symmetric key as well. The response to the request made to each entity is decrypted (lines 9–22) where integrity checks are also made. After this verification, the received contribution is extracted and added to the temporary list (line 23). When the requests to all entities have been processed, the system builds the final symmetric key of the group.

**Listing 1.** Symmetric key generation algorithm (in JavaScript).

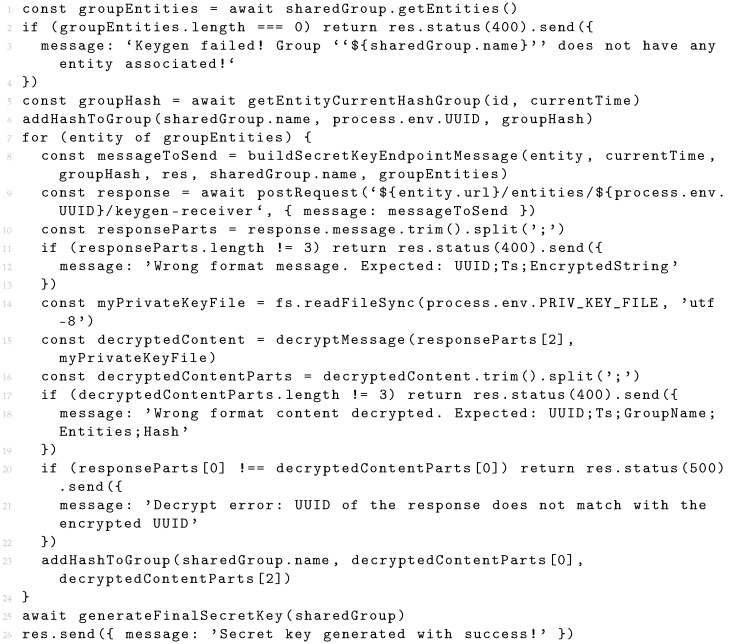



Similarly, an entity receiving a request for the construction of a shared symmetric key starts by performing some validations to ensure the integrity of the process. The identifier of the entity that made the request is checked against the identifier extracted from the received encrypted content, then the existence of a sharing group with the same name is verified, and finally, the entity that made the request is checked against the list of the entities that form the group. This verification is performed by comparing the list of identifiers of entities extracted from the request with the list of identifiers of entities obtained from its own database. After these validations, the process is similar to the one shown in Listing 1. The system starts by creating its own contribution, creates a temporary list to store the contributions of the other entities, and finally, requests contributions from the other entities in the group. However, there is one difference when compared to the process described in Listing 1. The system will only send requests to entities for which it does not yet have the corresponding contribution. This is controlled by checking its presence on the temporary list. At this point, the current entity has two contribution hashes: its own and the one received by the incoming request. Therefore, it will only be necessary to make requests to the remaining entities of the group. If this entity is at the end of the chain, then there will be no more requests because the temporary list of the group’s contribution hashes is already complete, and it is only necessary to send its contribution hash to the requesting entities.

### 3.3. Shared Index Update

The process of updating a shared index requires that an entity should be able to share part of its information with a restricted group of entities forming a sharing group in a confidential manner and guaranteeing the integrity of all exchanged information. The confidentiality of the information existing in the shared index is guaranteed by the use of the final symmetric key, described in the previous section. The integrity of this information is guaranteed through the use of the public–private key pairs of the entities belonging to the group.

The shared index was implemented using a remote Mongo database because this does not require a fixed structure and uses JavaScript Object Notation (JSON) structures, suitable to allocate the created indexes. The location of the index also requires an agreement between all participating entities. After agreeing on the index location, all entities update their group by creating a configuration. Of note is the fact that all information stored within this database is fully encrypted. The configuration of an index allows the user to define the periodicity and the update time instant for the execution of the index update process. Sharing groups and index configurations feature a one-to-one relation, meaning that for each sharing group, only one index can be defined and therefore only one configuration exists.

In order to create a new index configuration for a specific sharing group, three parameters are required to create a new configuration: (1) the address of the Mongo database where the index is to be located, (2) the index update period, and (3) the update time. The index update task is scheduled based on the update period and time specified by the user. The name of the scheduled task includes the task type concatenated with the entity identifier, thus avoiding name conflicts between the various tasks scheduled in the system. The index configuration requires the configuration of which information will be shared on the index and this information corresponds to the IoC-related information that the user wants to share with the other entities in the group.

Listing 2 presents the algorithm developed for updating a shared index. The system starts by identifying the group and its index configuration (lines 2–3) and verifies if the configuration is enabled (line 4). If so, the connection to the database is established (line 5). After establishing the connection, the system obtains the information to be uploaded to the index (lines 6–7) and the group symmetric key, which will be used to guarantee the confidentiality of the information in the index (line 9). After the system has all the data to perform the update process, the last check is made on the values that will be sent to the index. The system will verify if all values listed in the configuration are present in their MISP instance (lines 11–13). The user is able to add any IoC to the configuration of the index, but these must exist in the MISP database in order to be uploaded to the shared index. The structure of the adopted shared index is based on the concept of a reverse index, i.e., each key of the index has a list of values. The key corresponds to the trapdoor created over the indexed value and will be used to perform searches. The list of values corresponds to the result of the encryption of the identifier of the entity that performs the update process, concatenated with the signature of the generated trapdoor. This structure indicates that the result of a search process for a value (represented by a trapdoor in the index) forms a list of entities that have information about that same value. Initially, the system creates the trapdoor using the symmetric key of the shared group (line 14) and queries the index for any existing reference to this trapdoor (line 15). The existence of a reference indicates that some entity of the group has information about the trapdoor. Next, the trapdoor is signed (lines 16–17) using the private key of the entity that is updating the index. This signature will allow for the verification of the integrity of the trapdoors. The signature is encrypted together with the identifier of the entity performing the update (line 18). Finally, this encrypted content is inserted into the index (lines 19–24). If the trapdoor does not exist in the index, a new entry is created with the trapdoor and the encrypted content. If the trapdoor already exists, the encrypted content is added to the existing trapdoor.

**Listing 2.** Index update algorithm (in JavaScript).

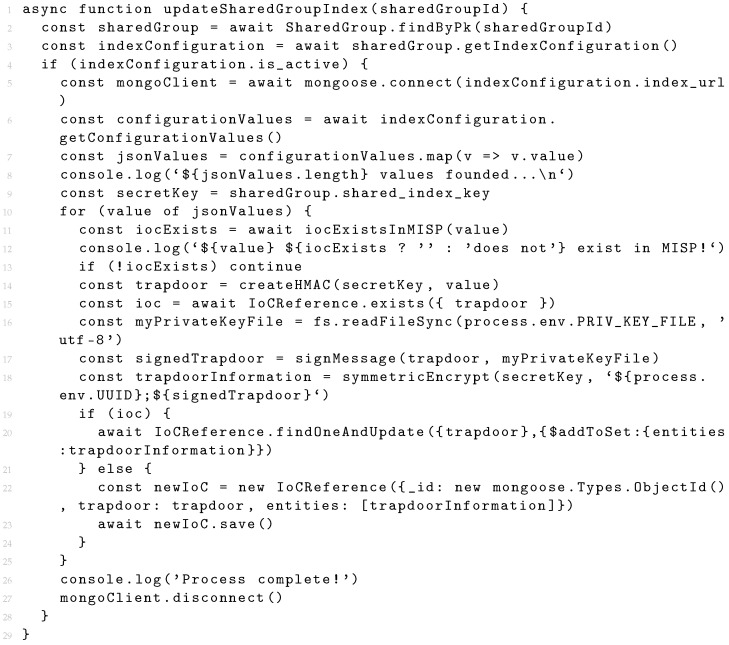



The shared index does not have any information in cleartext and thus only those possessing the symmetric key of the sharing group have access to the indexed data. In an eventual case where this key is exposed, a practical mitigation could be to delete all related index entries and generate a new key for the group. The index does not hold the information present in the MISP instances, only the references to the information, maintaining the security of this information.

## 4. Performance Evaluation

The prototype and API feature two processes that contribute the most to the overall performance: the process of generating the symmetric key of a shared group and the process of uploading data to the shared index. This section assesses these processes regarding their duration (also taking into account the duration of each phase of the process) and the usage of the CPU and RAM during the execution. For each test, 10 repetitions were performed, and the average values for the samples obtained per repetition were collected.

### 4.1. Symmetric Key Generation Evaluation

The objective of this test was to evaluate the duration of the symmetric key generation process of a shared group, taking into account the number of entities associated with that group. As mentioned in [2], the group’s symmetrical key is generated with the contribution of all the entities in the group. Thus, each entity generates a hash value and shares it with all the other entities. After sharing, all the entities have the same amount of hashes. The symmetric key is generated using the Secure Hash Algorithm (SHA) 512-bit hash function, having as input the hashes ordered in alphabetical order and truncating the output to 256 bits (SHA-512-256). Tests were performed for the following number of entities: 100, 200, 300, 400, and 500 entities and the process was divided into the following phases:Exchange hashes: the phase to exchange the hashes between all the entities present in the group (*exchangedHashes = numberOfEntities(numberOfEntities-1)*);Generate final key: the phase when the algorithm for generating the key is applied by all the entities in the group.

The obtained results from the number of entities tested are presented in Table 1. From these results, it is possible to verify that the “Generate final key” phase takes the least time, taking less than 0.1% of all the time spent in the process. On the other hand, the “Exchange hashes” phase occupies most of the time used in the process of generating the symmetric key of the shared group. The longer duration of the “Exchange hashes” phase can be explained by the fact that it depends on the communication between the different entities of the group. The more entities that belong to a shared group, the more requests will be made and, therefore, the completion time of this phase increases.

Measurements were made as to the behavior of the CPU and RAM during the execution of this process. The values shown correspond to the total percentage of the CPU and RAM used by the computer at the moment and not of the specific thread used for the execution of the processes. Figure 2a,b draw, respectively, the CPU and RAM behavior for the cases of 100 and 500 entities existing in a shared group. For the case of 100 entities (Figure 2a), the CPU presented an average usage of 6.11%, reaching a maximum value of around 28%. The RAM presented an average usage of 44.23%, reaching a maximum value of 44.89%. For the case of 500 entities (Figure 2b), the CPU presented an average usage of 4.57%, reaching a maximum value of around 86%. As regards the RAM, it presented an average usage of 42.85%, reaching a maximum value of 50.69%.

### 4.2. Shared Index Update Evaluation

The evaluation of the shared index update process is intended to assess the duration of the update process of the shared index, considering the number of IoCs being uploaded to the index. Considering that an element of the index corresponds to the trapdoor and its associated encrypted content (the signature of the trapdoor and the UUID of the entity that performs the upload), the following numbers of elements were used: 1000, 10,000, and 100,000 elements.

This evaluation was divided into the following five phases:Trapdoor creation: the duration of the Hash-based Message Authentication Code (HMAC) value generation, having as an input the secret key of the shared group and the IoC to be uploaded to the shared index;Trapdoor signature: the duration for the trapdoor to be signed using the private key of the entity that is executing the process;Trapdoor encryption: the duration for the creation of the encrypted content associated with the trapdoor;Trapdoor verification: a verification intended to check whether the shared index already contains the current IoC. If it already exists, the current IoC is dropped;Shared index upload: the duration for updating the MongoDB collection with the current IoC.

Table 2 presents the duration, in seconds, of each phase of the shared index update process, for 1000, 10,000, and 100,000 elements. Figure 3 draws the percentages of Table 2.

From the results obtained, it can be highlighted that the “Trapdoor creation” phase takes the least time in the whole process. On the other hand, the “Trapdoor verification” phase comprises the highest percentage of time spent on the entire index updating process. In the case of the update of 100,000 elements, this value increases more (from 39.92 to 2607.60 s). Updating the index is fast with 1000 elements because the times of each phase are almost negligible (a total duration of approximately 4 s). For the case of 10,000 elements, the process increases its total time to approximately 67 s, starting to show the trend that the “trapdoor verification” phase corresponds to the largest percentage of time spent (approximately 40 s, more than half of the total time). For the case of 100,000 elements, the total time increases, based on the “trapdoor verification” phase accounting for approximately 90% of the total time spent on the process. Thus, the “Trapdoor verification” phase becomes a big burden on the overall duration of the process. At runtime, each trapdoor corresponds to one or two requests to the shared index on the MongoDB server, depending on its presence on the server. The first request consists of checking whether the trapdoor exists in the shared index. The second request consists of inserting the trapdoor into the shared index if it does not already exist. Summing up, in the worst-case scenario, the number of possible connections corresponds to twice as many candidate trapdoors to be inserted in the shared index.

As in the process of the symmetric key generation, measurements were also made of the CPU and RAM behavior during the execution of this process. Figure 4a–c present the total percentage of the CPU and RAM used per second, respectively, for the cases of 1000, 10,000 and 100,000 elements. For the case of 1000 elements (Figure 4a), the CPU had an average occupation of 14.37%, reaching a maximum value of around 35%. The RAM had an average occupation of 40.60%, reaching a maximum value of 49.35%. For the 10,000 elements case (Figure 4b), the CPU had an average occupation of 13.32%, reaching a maximum value of 35%. In this case, the RAM had an average occupation of 44.26%, reaching a maximum value of 50.14%. Finally, for the 100,000 elements case (Figure 4c), the CPU had an average occupation of 13.93%, reaching a maximum value of 100%. Regarding the RAM, it had an average occupation of 46.19%, reaching a maximum value of 53.06%.

After analyzing the overall performance results in the shared index update evaluation, an effort was made to improve the update process in order to reduce the connections with the MongoDB server, thus reducing the total time duration of the process.

## 5. Proposed Improvements

In order to optimize the results in the shared index update process, a caching system was adopted to enable the reduction in the duration of the trapdoor verification phase. The cache is responsible for storing only the trapdoor value corresponding to the IoC that is added by the system to the shared index. In the first iteration of the shared index update process in the trapdoor verification phase, the system needs to perform the query on the index to check if the trapdoor exists because the cache does not contain any stored value. At the end of the iteration, the trapdoors identified as valid to be inserted in the shared index are also added to the cache. In the following iterations in which the process is executed, the system first queries the cache, only querying the shared index if the trapdoor is not available in the cache (e.g., a new IoC that appears in the MISP database). Thus, the process is faster because querying data in memory takes less time than querying data between different servers.

Table 3 shows the duration, in seconds, for each of the phases of the shared index update process using the caching system. Table 4 presents the variation for each phase of the shared index update process using the cache, compared with no cache. These results allow verifying that the trapdoor verification phase decreases mainly in the trapdoor phase.

Figure 5 presents the weight (percentage) of the phases involved with no cache (NC) and with the cache (C). Table 5 presents the total duration of the shared index process with no cache (NC) and with the cache (C).

From these results, it is possible to verify that the process of updating the index became much faster using the caching system. Comparing these values with the values presented in Section 4.2, the most significant difference is in the case of 100,000 elements, where the process went from approximately 2920 s to approximately 302 s, which corresponds to a decrease of approximately 2618 s relative to the first results. This decrease is due to the difference in the results in the trapdoor verification phase. Without the use of the cache, this phase corresponded to the highest percentage of time in the process, while through the caching system, this value moved to the middle of the table. For the case of 1000 elements, the trapdoor verification phase corresponds to the lowest duration of all the phases, while in the case of 10,000 elements, this phase occupied the third lowest of all the phases in the process.

## 6. Security Analysis

The peering validation process shown in [2] consists of a variant of the well-known Needham–Schroeder Public Key (NSPK) protocol [32] and is used only to confirm that the communicating parties are correctly configured. This is performed by exchanging newly created random values (or nonces). This exchange is encrypted using the public key of the other party, imposing confidentiality, and timestamps are added to prevent replay attacks. The used keys are assumed to be securely exchanged, mainly because of it being an offline and by-hand procedure, due to the military application of the solution.

Nevertheless, the Automated Validation of Internet Security Protocols and Applications (AVISPA) [33] tool was used, and it allows for the automated validation of the security protocols described in High-Level Protocol Specification Language (HLPSL) [34]. The referred peer validation process was then described in the HLPSL in such a way that the nonces generated by both peers could have their secrecy verified. Appendix A shows the HLPSL specification of the peering validation. It comprises two roles (Alice and Bob), each representing an entity. For simplification, the exchanged messages were reduced to the encrypted parts of the peering protocol, the reasoning being that the content not encrypted was deemed as known by all. Considering that the AVISPA tool does not support the timestamp concept, the timestamps were not included. The tool reported the protocol as secure when executing all of the supported back-ends, meaning that the secrecy of the exchanged nonces was confirmed.

## 7. Conclusions

The MISP enables the control of cyberthreat sharing and facilitates security incident handling as a community by connecting and sharing information concerning attacks or threats. To impose greater control over shared cyberthreat information, a previous solution using Symmetric Searchable Encryption was proposed by the same authors.

This paper describes the prototype implementation of the key functionalities of the previous solution and evaluates its performance. After the performance evaluation, an enhancement opportunity was detected that consisted of improving the shared index update process by creating a local cache memory of information already inserted into the index, leading to the reduction in queries to the shared index. This enhancement resulted in the shortening of the duration of the update process from around 2922 s to around 302 s, when considering a shared index with 100,000 elements.

A security analysis was also performed using the AVISPA which confirmed that the implementation protocol can be considered secure when executing all of the supported back-ends, thus confirming the secrecy of the exchanged nonces.

As future work, one direction could be to extend the current encrypted search mechanism to support more than one IoC at a time, possibly combining them using logical expressions.

## Figures and Tables

**Figure 2 sensors-23-00914-f002:**
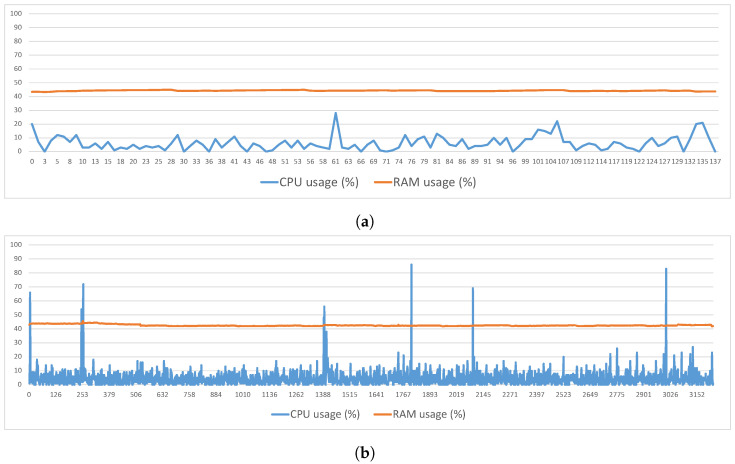
Symmetric key generation—RAM and CPU usage. (**a**) RAM and CPU usage—100 entities, (**b**) RAM and CPU usage—500 entities.

**Figure 3 sensors-23-00914-f003:**
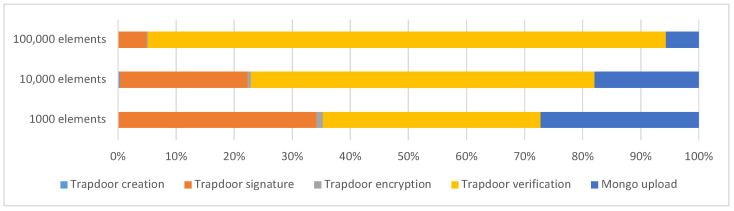
Shared index update duration—chart (in seconds).

**Figure 4 sensors-23-00914-f004:**
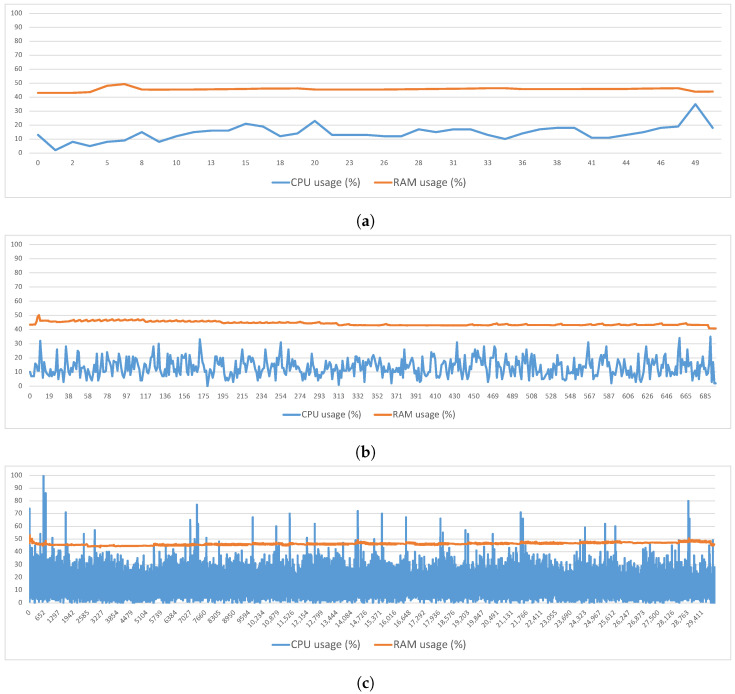
Shared Index Update—RAM and CPU Usage. (**a**) 1000 elements, (**b**) 10,000 elements, (**c**) 100,000 elements.

**Figure 5 sensors-23-00914-f005:**
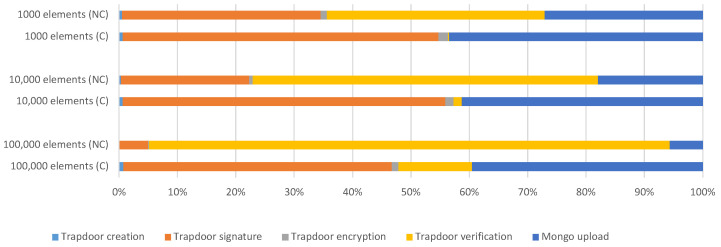
Shared index update duration per stages—with no cache (NC) and with cache (C), in seconds.

**Table 1 sensors-23-00914-t001:** Symmetric key generation duration (in seconds) for 100, 200, 300, 400, and 500 entities.

Entities	100	200	300	400	500	%
Exchange hashes	12.76	51.17	115.30	205.99	321.40	>99.9%
Generate final key	0.01	0.03	0.07	0.13	0.22	<0.1%
Total duration	12.77	51.20	115.38	206.13	321.68	100.00%

**Table 2 sensors-23-00914-t002:** Shared index update duration for 1000, 10,000, and 100,000 elements (in seconds and percentages).

	1000 Elements		10,000 Elements		100,000 Elements	
Trapdoor creation	0.02	0.51%	0.21	0.32%	2.54	0.09%
Trapdoor signature	1.36	34.00%	14.82	21.97%	142.90	4.89%
Trapdoor encryption	0.04	1.09%	0.39	0.58%	3.85	0.13%
Trapdoor verification	1.49	37.31%	39.92	59.17%	2607.60	89.22%
Mongo upload	1.08	27.09%	12.12	17.97%	165.68	5.67%
**Total duration**	**4.00**	100.00%	**67.46**	100.00%	**2922.56**	100.00%

**Table 3 sensors-23-00914-t003:** Shared index update process duration with cache—results (in seconds and percentages).

	1000 Elements		10,000 Elements		100,000 Elements	
Trapdoor creation	0.02	0.64%	0.16	0.64%	2.14	0.70%
Trapdoor signature	1.30	54.08%	13.96	55.26%	139.29	46.02%
Trapdoor encryption	0.04	1.62%	0.35	1.39%	3.49	1.15%
Trapdoor verification	0.01	0.20%	0.35	1.40%	38.08	12.58%
Mongo upload	1.05	43.60%	10.44	41.30%	119.65	39.53%
**Total duration**	**2.41**	100.00%	**25.27**	100.00%	**302.65**	100.00%

**Table 4 sensors-23-00914-t004:** Shared index update process duration with cache—variation in results compared with no cache (in seconds and percentages).

	1000 Elements		10,000 Elements		100,000 Elements	
Trapdoor creation	0.00	0.31%	−0.05	0.12%	−0.40	0.02%
Trapdoor signature	−0.06	3.69%	−0.86	2.05%	−3.16	0.14%
Trapdoor encryption	0.00	0.30%	−0.04	0.09%	−0.36	0.01%
Trapdoor verification	−1.49	93.31%	−39.56	93.76%	−2569.51	98.08%
Mongo upload	−0.04	2.39%	−1.68	3.99%	−46.03	1.76%
**Total improvement**	−1.59	100.00%	−42.20	100.00%	−2619.91	100.00%

**Table 5 sensors-23-00914-t005:** Total duration of shared index update process duration with and with no cache (in seconds).

Total Duration	1000 Elements	10,000 Elements	100,000 Elements
without cache	4.00	67.46	2922.56
with cache	2.41	25.27	302.65

## Data Availability

Data sharing not applicable.

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
