# Peer review of "On the Performance of Secure Sharing of Classified Threat Intelligence between Multiple Entities"

_sensors, 2023, doi:10.3390/s23020914_

Round 1
Reviewer 1 Report
This paper describes a system for sharing security incidents. While the underlying cryptography is not completely novel, the key generation algorithm and proof of concept will be interesting to practitioners. I do believe, that a protocol diagram describing for the key generation process would be helpful to readers.
There are additionally some minor errors in the presentation:
* Line 2: I would like to have seen the MISP acronym spelled out.
* Line 31: "based on" should be replaced with "a" SSE is a form of searchable encryption -- not based on it.
* Line 93: "tentative" is the incorrect word -- do the authors mean "techniques"?
* Line 136: LaTeX macro typo texitBig O should be \textit{Big O}
* Captions for code listings should mention that they are in JavaScript.
* Line 292: "symmetrical" should be "symmetric"
Author Response
Please see attached report.

Reviewer 2 Report
This paper describes a prototype implementation of two key functionalities used for multiple entities sharing information with each other: the symmetric key generation of a sharing group and the functionality to update a shared index. Also, these functionalities are evaluated regarding their performance and enhancements. In addition, this paper showed that the duration of the update process was shortened from around 2922 seconds to around 302 seconds, when considering a shared index with 100,000 elements. Experimental results have demonstrated the effectiveness of the proposed method in confirming the secrecy of the exchanged nonces.
The paper describes the prototype implementation of the previously proposed system considering multiple elements exchanging information with each other and evaluates the performance of this system based on a searchable encryption technique. The evaluation takes into account the two key processes namely the process of generating the symmetric key of a shared group and the process of uploading data to the shared index (MongoDB collection). The two processes are evaluated in duration and RAM and CPU usage during process execution. The paper also proposes and describes an index update process improvement by including a cache system to reduce the duration of the update process.
Overall, this work is quite interesting and innovative. The paper is easy to follow.
A few comments:
1. Can summarize the contributions of the paper.
2. Please check and supplement the full names of some abbreviations in the paper, such as "MISP" in the abstract. In addition, in the part of “Abbreviations”, the abbreviations are not complete, please supplement appropriately.
3. Can show the detailed software and hardware settings of this paper in the form of a table.
4. In Section 3, please add a flowchart to show the implementation process of the prototype implementation in more detail.
5. Need to use pseudo code for algorithm description.
6. Most of the references in this paper are relatively old. Please add references in recent years appropriately.
7. Numerous experimental results have proved the effectiveness of the proposed method. It would be great if the paper could compare with related work in secure sharing of classified threat intelligence between multiple entities.
Author Response
Please see attached report.
